# Early Diagnosis of Oral Mucosal Alterations in Smokers and E-Cigarette Users Based on Micronuclei Count: A Cross-Sectional Study among Dental Students

**DOI:** 10.3390/ijerph182413246

**Published:** 2021-12-16

**Authors:** Anca Maria Pop, Raluca Coroș, Alexandra Mihaela Stoica, Monica Monea

**Affiliations:** 1Faculty of Medicine, George Emil Palade University of Medicine, Pharmacy, Science, and Technology of Târgu Mureș, 540139 Tirgu Mures, Romania; ancapop98@yahoo.com; 2Faculty of Dental Medicine, George Emil Palade University of Medicine, Pharmacy, Science, and Technology of Târgu Mureș, 540139 Tirgu Mures, Romania; raluca_coros@yahoo.com; 3Department of Odontology and Oral Pathology, George Emil Palade University of Medicine, Pharmacy, Science, and Technology of Târgu Mureș, 540139 Tirgu Mures, Romania; alexandra.stoica@umfst.ro

**Keywords:** cytodiagnosis, oral health, micronucleus assay, cigarette smoking, e-cigarette

## Abstract

The presence of micronuclei in oral epithelial cells is considered a marker of genotoxicity, which can be identified using exfoliative cytology. The aim of this study was to investigate cytotoxic damage through the evaluation of micronuclei in the oral mucosa of smokers and e-cigarette users compared to nonsmokers. We obtained smears from the buccal mucosa of 68 participants divided in 3 groups (smokers, e-cigarette users and nonsmokers), which were further processed with Papanicolaou stain. The frequencies of micronuclei and micronucleated cells were recorded and statistically analyzed at a level of significance of *p* < 0.05. The mean micronuclei values per 1000 cells were 3.6 ± 1.08 for smokers, 3.21 ± 1.12 for e-cigarette users and 1.95 ± 1.05 for nonsmokers. The mean values of micronucleated cells per 1000 cells were 2.48 ± 0.91 for smokers, 2.39 ± 1.07 for e-cigarette users and 1.4 ± 0.68 for nonsmokers. Smokers and e-cigarette users had significantly higher values of micronuclei and micronucleated cells compared to nonsmokers, but there were no significant differences between smokers and e-cigarette users. We concluded that the micronuclei count can be used as an early indicator for alterations of oral mucosa and exfoliative cytology represents an accessible tool which could be applied for mass screening.

## 1. Introduction

Smoking represents the most common cause for oral cancer and was identified as the second most significant risk factor responsible for global death; its by-products such as polycyclic hydrocarbons and nitrosamine have mutagenic and carcinogenic properties, as these compounds infiltrate into the oral mucosal cells and might induce cellular changes [1]. Adolescence has been associated with a high risk of substance use, among which tobacco smoking is the most frequent. The majority of smokers had their first cigarette or were already addicted before they turned 18 and compared to adults, young people need fewer cigarettes and less time to become nicotine addicted [2]. E-cigarettes were developed in 2003 in an effort to assist smoking cessation and to reduce the harm effects of tobacco and other substances present in conventional cigarettes [3]. These devices consist of an atomizer for heat generation, a battery with fixed voltage and a liquid reservoir containing a mixture of volatile compounds alongside with different amounts of nicotine, propylene glycol, glycerol, water, flavoring agents and dyes [4,5,6]. There are limited data regarding the impact of e-cigarettes on adolescents and young adults, but until now it is clear that their use is not restricted to current smokers and due to high youth awareness, these devices are becoming more popular [7]. In USA, between 2010–2013 the use of e-cigarettes in adults raised from 1.8% to 13% [8]; the highest prevalence was reported among young adults aged 18–24 years old and among former smokers but also individuals who never smoked before [9]. However, in spite of the continuous increase in the use of e-cigarette among young people, information about its possible negative consequences on the oral and overall health is still limited [10].

It is well known that smoking conventional cigarettes is associated with poor dental health, with inflammation of oral tissues leading to gingival and periodontal disease [4,11], but only few studies focused on the correlations between e-cigarettes and oral health status. An association between e-cigarette use and tooth fractures in adolescents has been suggested [12] but also the observations that in comparison to smokers, individuals that use e-cigarettes and those who never smoked have less periodontal inflammation [11,13]. A recent review [14] showed that some studies reported that e-cigarette users exhibit higher levels of proinflammatory cytokines, higher plaque index and more frequent periodontal disease compared to nonsmokers [15,16], but others did not confirm this observation [17,18]. Moreover, most of the studies focusing on the effects of e-cigarettes either compared their users with conventional smokers, or with nonsmokers and only a few studies compared these two types of nicotine users with controls (nonsmokers) [19,20]. Even so, studies who included all the three groups mentioned above selected patients older than 45 years, with long-term exposure to nicotine [19] or with already present clinical alterations or lesions of the oral mucosa [20].

The conflicting data raised many concerns upon the effects of these devices on the oral health [21]. The association between smoking and alcohol increases the risk of occurrence for both oral premalignant and cancerous lesions, as the cariogenic compounds present in the smoke are dissolved in alcohol, which also determines vasodilation and enhances the absorption of these substances by the oral mucosa [22,23]. All forms of tobacco use can cause cancer, therefore early detection and treatment of oral premalignant lesions are of utmost importance in reducing the morbidity, mortality and costs associated with the treatment of oral malignancies [22].

Exfoliative cytology allows the early identification of morphological changes in the oral mucosa; it is a simple and non-invasive technique considered to be a valuable adjuvant in the diagnosis of premalignant lesions, allowing appropriate preventive measures against the development of oral cancer [24]. It is also suitable for the follow-up of premalignant lesions, because it avoids the use of repeated biopsies, which present disadvantages such as the invasive character, esthetic concerns, risk of infection or damage to normal tissues [25]. The advancements in this method facilitated the evaluation of genotoxic exposure using different parameters such as cellular and nuclear diameters, nuclear shape and discontinuity or assessment of micronuclei (MN). The latter are extracellular cytoplasmic bodies that are formed during anaphase from chromosomal fragments. Several studies showed that there is a correlation between the frequency of MN and severity of the genotoxic damage, these being further used for grading the lesion [26,27]. Therefore, MN count is regarded as an indicator of chromosomal aberrations and is useful in detecting early-stage carcinogenesis [28,29].

As there are limited data regarding the presence of MN in clinically normal oral mucosa in different types of young nicotine users compared to nonsmokers, the aim of our study was to evaluate based on exfoliative cytology, in a group of young adults, the frequency of MN in the oral epithelial cells of smokers, e-cigarette users and never smokers.

## 2. Materials and Methods

### 2.1. Study Participants and Selection Criteria

After the approval of the study by the Ethics Committee of our university, all students from the Faculty of Dental Medicine were invited to participate in this research. An e-mail was sent in February 2021 containing the aim of the study and a questionnaire with inclusion and exclusion criteria. It was mentioned that the enrollment is voluntary and the data will remain anonymous. Participants were defined as nonsmokers if they had no history of smoking, as smokers if they consumed at least 10 conventional cigarettes/day in the past 12 months and as e-cigarette users if they recalled current consumption of e-cigarettes daily in the past 12 months, with no conventional cigarette smoking in the last 3 months. As exclusion criteria we used the following conditions: consumption of other forms of tobacco (hand-made cigarettes, country-style cigarettes, cigars and pipes or tobacco chewing), dual users (current simultaneous use of conventional and e-cigarettes), chronic use of alcohol (>7 units/week for women and > 14 units/week for men) or other drugs, orthodontic treatment in progress, alteration of the oral mucosa or previous diagnosis of oral premalignant lesions, systemic diseases (diabetes, autoimmune diseases, hematological disorders). A total of 68 participants, who agreed to take part in the research and met the inclusion and exclusion criteria, were enrolled in three groups: A-25 conventional cigarette smokers, B-23 e-cigarette users and C-20 nonsmokers.

### 2.2. Exfoliative Cytology Sampling Technique

The participants were asked to rinse the oral cavity with tap water and the excess of saliva and debris were gently wiped off with a cotton pellet. A thorough examination was performed by a single investigator, who was not aware to which study group the participant belonged to. Buccal cells were collected using a wooden spatula and then transferred on glass slides. For each participant two glass slides (one from each cheek) were prepared and fixed with 95% ethanol. The specimens were further processed for histopathological evaluation using Papanicolaou (Pap) stain.

### 2.3. Histopathological Protocol

For the Pap stain, the slides were introduced in 1% acetic acid (10 dips) and then treated with Hematoxylin preheated 60 °C. They were washed in tap water and dipped in 1% acetic acid. They were then treated with Orange Green-6, followed by 1% acetic acid (10 dips) and in Eosin Azure-50. Again, they were dipped by 1% acetic acid, methanol (10 dips), and xylene (10 dips). The slides were coded, then randomized and scored by one examiner in a single-blind manner (the examiner was not aware to which group the slides belonged).

### 2.4. Micronucleus Analysis

All slides were viewed under an optical microscope (Zeiss, Oberkochen, Germany) at 10x magnification for screening and 40x magnification for MN counting. For each participant, 1000 cells were evaluated and the frequency of MN and micronucleated cells (MNC) per 1000 cells were recorded by a single examiner, who demonstrated good intraobserver reliability, with an intraclass correlation coefficient (ICC) of 0.84 (0.82–0.87). Only cells which met the following criteria were included in the analysis: presence of intact cytoplasm and relative flap cell position on the slide, little or no overlap with adjacent cells, little or no debris, normal and intact nucleus with smooth and distinct nuclear perimeter without nuclear overlapping.

The previously published criteria by Tolbert et al [30] used for identifying MN were applied: texture similar to the nucleus, normal smooth perimeter, less than 1/3 the diameter of the nucleus, absence of overlap or bridge to the nucleus. The criteria for excluding cells for MN assessment were also followed: cells with two nuclei, degenerating or dead cells, presence of MN-like structure connected with the main nucleus with a bridge.

### 2.5. Statistical Analysis

Descriptive statistics was obtained using Microsoft Excel (Microsoft, Redmond, WA, USA) and ICC with 95% confidence interval was calculated by SPSS Statistics version 25 (IBM Corporation, Armonk, NY, USA). Data were statistically analyzed using GraphPad Prism 7 (GraphPad Software, San Diego, CA, USA) and were summarized as mean ± standard deviation (SD) for continuous variables and as absolute numbers and percentages for categorical variables. Kolmogorov-Smirnov test was used for assessing normal distribution, and then data were compared by one-way analysis of variance (ANOVA) or Kruskal-Wallis test, as appropriate. Post-hoc analysis was conducted using Tukey Kramer or Dunn’s multiple comparison test, as appropriate and also in particular situations with Mann-Whitney test for assessing trends. Categorical variables were analyzed using Chi-square test. The level of statistical significance was set at a value of *p* < 0.05 (two-tailed).

## 3. Results

The demographic data of the participants are summarized in Table 1. The groups were considered homogenous, as there were no statistically significant differences regarding gender (*p* = 0.4119) and age (*p* = 0.1553) distribution. 

Examples of representative histopathological aspects are presented in Figure 1 and Figure 2.

The distribution of MN and MNC values, expressed as median and interquartile range in the three study groups is presented in Figure 3.

There was a statistically significant difference between the three groups regarding MN (*p* < 0.0001) and MNC count (*p* = 0.0007). (Table 2)

The results of the post-hoc Dunn’s analysis are presented in Table 3. The values of both smokers (Group A) and e-cigarette users (Group B) were significantly different from the values of nonsmokers (Group C) regarding MN and MNC count. Conversely, between smokers (Group A) and e-cigarette users (Group B) these two parameters showed no statistically significant differences. However, in order to obtain the exact value of *p* regarding the comparison between Groups A and B we also performed the Mann-Whitney test as post-hoc analysis and obtained a *p* value of 0.2207 for MN count and 0.7953 for MNC count, respectively. These values were interpreted as non-suggestive for trends (*p* > 0.1).

## 4. Discussion

The aim of this study was to determine whether smokers or e-cigarette users exhibit cytotoxic damage of the buccal mucosa in the absence of any significant clinical manifestation noticeable to the individual, which could be a false indicator of wellbeing. The early diagnosis of oral lesions has become an important objective, as it has an immense impact on the successful treatment of these patients [31]. We selected students without chronic use of alcohol, dual use of conventional and e-cigarettes or systemic diseases which are known to influence the oral mucosa. In the e-cigarette users group we included students who previously smoked conventional cigarettes but replaced them with e-cigarettes for at least 3 months, as there were no participants who never experienced conventional smoking before.

The frequency of MN was proved to be an accurate hallmark of carcinogenesis, having a higher value in cancerous compared to precancerous lesions [32]. Their quantitative assessment can be considered an indicator of genetic damage, because in the majority of cases cells with MN which enter mitosis produce daughter cells with higher number of MN [23,33]. Moreover, the evaluation of other parameters gave contradictory results, as nuclear abnormalities specific for apoptosis (condensed chromatin, karyorrhexis and pyknosis) were identified less frequently in tumoral compared to normal cells [32]. However, as most of the research conducted focused on patients with oral lesions or with long-term exposure to genotoxic agents, the novelty of our study is represented by the investigation regarding the presence of MN in clinically normal oral mucosa and in a young age group with relative short exposure to carcinogens. In apparently healthy smokers, several studies have reported morphological alterations such as high rate of epithelial cell proliferation, the presence of MN, increasing number of keratinized cells and an altered ratio between nuclear and cytoplasmic volumes [34,35]. 

Based on a recent Cochrane analysis [36], oral cytology showed 90% sensitivity and 94% specificity in detecting oral lesions. Its performance was better in comparison with vital staining and light-based techniques, as these can erroneously provide more false positive results due to lower specificity and are difficult to interpretate by the non-expert clinician. Although the diagnosis of any oral lesion still needs a biopsy for confirmation, oral cytology represents a promising technique based on the following arguments: the high sensitivity supports its application for the screening of small, homogenous lesions, which do not exhibit clinical features suggestive for malignancy, non-invasive character, low cost and ability to provide a quantitative analysis of reliable genotoxicity. [36,37] Therefore, oral cytology was found to be the most precise method for oral lesions diagnosis, after the surgical biopsy, which make it a large-scale applicable technique for screening and early diagnosis of malignant lesions [36] In our study we selected the buccal mucosa for collection site, as cells located in this area are suitable for the assessment of any changes developed in relation to smoking. Several collection instruments have been proposed for harvesting oral epithelial cells, such as wooden or metal spatula and different types of cytobrushes. In our study we used wooden spatulas, as they are efficient and easily accessible, although cytobrushes have been considered to provide better homogeneity of the smears and to facilitate cell collection from all layers of the mucosa [38]. In the aforementioned Cochrane meta-analysis the sensitivity in detecting oral mucosal abnormalities reported values of 93% for smears collected by scraping (spatula) and 91% for brush, respectively. The specificity showed a value of 92% for scraping and 94% for brush [36].

Among commonly used stains for MN such as Pap, Feulgen and Giemsa, we used Pap stain, as it is considered the most effective staining technique for cytological assessment, which enables the visualization of clear nuclear and cytoplasmatic characteristics and emphasizes differences between cells located in various layers of the epithelium [39]. However, Feulgen stain was also proposed for assessing MN count due to its DNA specificity, but it was regarded as expensive, with more complicated fixation and staining process and it also requires additional compounds for the contouring of the cytoplasmatic borders. Therefore, based on the acceptable accuracy regarding MN evaluation, Pap stain may be used for routine screening and samples with potential abnormalities can be further processed with DNA-specific stains [40].

However, there are several factors which can interfere with the correct estimation of MN count, such as bacteria normally present in the oral cavity, but microorganisms usually have smaller size, various shape and are located also between oral cells. Moreover, small dye granules need to be differentiated from MN based on the more intense staining and different refractility [41].

Several normal ranges for MN frequency have been previously proposed, between 0.05–11.5 MN/1000 cells, with the majority of studies reporting values between 0.5–2.5 MN/1000 cells [42]. The values of MN obtained in our study for the nonsmokers group fall within the range of 0.5–2.5 MN/1000 cells, but the other groups exposed to carcinogens exhibited higher values, however still fitting in the normal range. A possible explanation for this can be the young age of the participants, the relative decreased number of package-years in the group of conventional smokers and the fact that exfoliative cytology can sometimes underestimate the MN count due to the time needed for the migration of cells from basal to superficial layers of the oral epithelium. We found that more cells showing MN were present in specimens from smokers and e-cigarettes users compared with nonsmokers, similar to data reported by other studies [29,43]. Furthermore, a systematic review of clinical studies published in 2018 found a higher frequency of MN in exfoliated cells of smokers compared to non-smokers and concluded that this is associated with cytotoxic and genotoxic effects [44]. In an effort to exclude the bias caused by a normal aspect of only occasionally exposed to carcinogens tissues, we selected participants with consistent, continuous and recent smoking or e-cigarette use, because in the absence of a persistent exposure only one sample may fail in detecting temporary cellular abnormalities [45]. Oral epithelial cells have a relatively decreased turnover [34]. However, in our study we also found an increased turnover, which may indicate the toxicity of several compounds released from tobacco. Smoking usually causes cellular irritation which predisposes to accelerated proliferation and altered cellular morphology [22].

E-cigarettes have become more popular in the last years as an alternative to conventional cigarettes and data show that the decrease of conventional smokers or dual users is accompanied by an increase in exclusive e-cigarette users. Although considered safer in comparison to conventional cigarettes, e-cigarettes have been associated with health risks, including cardiac and pulmonary diseases, but also with nicotine addiction leading to a tendency of experiencing other forms of tobacco [46,47]. While conventional smoking is recognized as an important risk factor affecting the oral health, there is still a debate in the scientific literature regarding the possible risks of e-cigarette use. Several studies reported milder oral symptomatology in e-cigarette compared to conventional cigarette users [48], but others found more frequent oral lesions, burns or inflammation compared to former or nonsmokers [20]. Therefore, recent data support the need for correct information about health-related risks before the decision of using these products, which are highly prevalent in the young population [47].

Several studies have observed that smoking initiation is common between 11 and 13 years of age, a time when adolescents may become very easily addicted to nicotine. Moreover, it has also been reported that some electronic nicotine delivery devices may have a more aggressive addictive potential compared to conventional cigarettes [49]. The trend of e-cigarettes was noticed also in Romania, where students use these devices 2.4 times more frequently than adults [50]. Due to the insufficient data regarding the full spectrum of risks associated with e-cigarettes, their use should be monitored especially in young people. Proper information is of utmost importance as preventive strategies need to be supported by solid scientific evidence, which should be easily accessible and understandable by large groups of the population.

## 5. Conclusions

Nicotine containing products have a negative influence on the oral health, which emphasizes the importance of correct information regarding the side effects of both conventional and e-cigarette use. Clinically healthy young tobacco smokers and e-cigarette users presented an increased number of MN in the oral epithelial cells, compared to nonsmoking individuals, suggesting the presence of cytologic changes. The MN count can be used as an early indicator for susceptibility to alterations of oral mucosa and carcinogenesis. Exfoliative cytology is a noninvasive, rapid and cost-effective method that could be used for mass screening.

## Figures and Tables

**Figure 1 ijerph-18-13246-f001:**
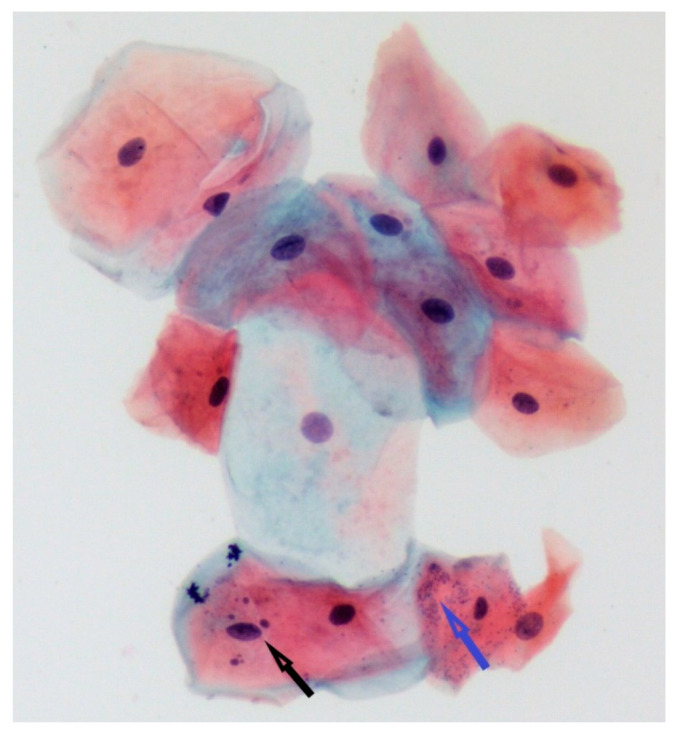
Specimen from conventional cigarette smoker. One cell with 4 MN was observed in a field with 15 epithelial cells. MN were identified based on their well-defined shape, size (less than 1/3 of the nucleus), similar color and refractility with the nucleus and absence of any connection to it (bridge) (black arrow). Microorganisms are recognizable as small, numerous bodies inside and between epithelial cells (blue arrow). (Pap stain, ×20).

**Figure 2 ijerph-18-13246-f002:**
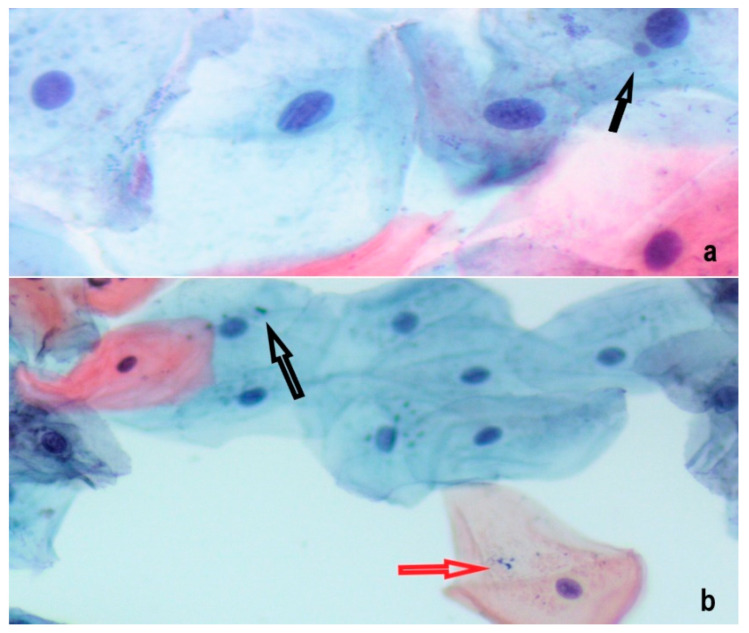
Epithelial cells containing 1 MN (black arrows) from (**a**) e-cigarette user (Pap stain, ×40) and (**b**) nonsmoker (Pap stain, ×20). Beside microorganisms, dye granules can be distinguished due to intense staining (red arrow).

**Figure 3 ijerph-18-13246-f003:**
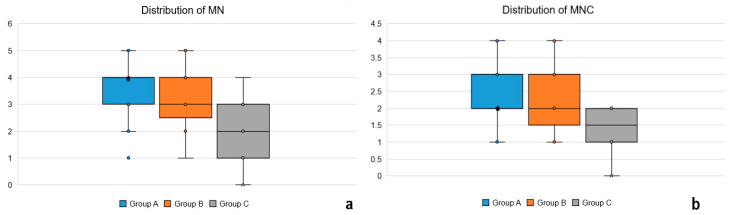
(**a**) Boxplot illustrating the distribution of MN among the three study groups; the median of Group A was identical with the value of quartile 3; (**b**) Boxplot illustrating the distribution of MNC among the three study groups; the median of Group A was identical with the value of quartile 1.

**Table 1 ijerph-18-13246-t001:** Demographic characteristics of the study groups.

Study Group	Gender n (%)		Age Mean ± SD
	Male	Female	
Group A	14 (60.41%)	11 (39.59%)	22.36 ± 1.41
Group B	15 (65.2%)	8(34.8%)	21.52 ± 1.65
Group C	9 (45%)	11 (55%)	21.7 ± 1.59
*p* value	0.4119 *		0.1553 **

* No statistically significant difference based on Chi-square test. ** no statistically significant difference based on ANOVA.

**Table 2 ijerph-18-13246-t002:** Mean values of the MN and MNC count in the study groups.

Group	MN Count Mean ± SD	MNC Count Mean ± SD	Number of Samples Containing MNC
Group A	3.6 ± 1.08	2.48 ± 0.91	25
Group B	3.21 ± 1.12	2.39 ± 1.07	23
Group C	1.95 ± 1.05	1.4 ± 0.68	18
*p* value	<0.0001 *	0.0007 *	

* Statistically significant difference based on Kruskal-Wallis test.

**Table 3 ijerph-18-13246-t003:** Results of the Dunn’s multiple comparison test.

Comparison	MN Count		MNC Count	
	Mean Rank Difference	*p* Value	Mean Rank Difference	*p* Value
Group A vs. B	6.453	>0.05	2.327	>0.05
Group A vs. C	25.015	<0.001 *	20.24	<0.01 *
Group B vs. C	18.562	<0.01 *	17.913	<0.01 *

* Statistically significant difference based on the adjusted value of *p* according to Dunn’s multiple comparison test.

## Data Availability

All relevant data presented in this study are contained within the article.

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
