# Peer review of "Early Diagnosis of Oral Mucosal Alterations in Smokers and E-Cigarette Users Based on Micronuclei Count: A Cross-Sectional Study among Dental Students"

_ijerph, 2021, doi:10.3390/ijerph182413246_

Round 1
Reviewer 1 Report
General comments:
Well written and designed study with significant implications for e-cigarette users. Data analysis and presentation needs to be improved to fully understand implications of the study.
Major comments:
- Please provide distribution of the number micronucleated cells per group.
- A boxplot graphically depicting the numerical data through their quartiles, would be useful.
- An average # micronucleated cells can be misleading plus from a perspective of increasing cancer risk, the number of users with micronucleated samples will had value to the study.
- Figure 3 and Table 2 are just different representations of the same data. Recommend choosing one format.
Author Response
Dear Reviewer,
In the attachment you may find our response to your observations.
Yours faithfully,
Monica Monea

Reviewer 2 Report
1. The difference and novelty between the existing study and this study is lacking in the introduction
- It is necessary to highlight the limitations of existing studies or differences due to new attempts
- Lack of diagnostic accuracy of the new test method
- It is important to mention the diagnostic accuracy compared to the golden standard
2. Statistical analysis method
- It is questionable whether normality is satisfied because the number of samples is not the same
- Comparison with existing diagnostic methods is necessary
- Analyst's reliability evaluation is required
- In-inspector and inter-inspector evaluation results are required
3. Results
-Table 1 has no meaning- Only the contents are described after deletion
-Picture 1: It should be changed to mainly describe the identification according to the color of the arrow.
Table 2: No post-analysis indication - Must be marked with post-analysis text after Man Whitney U-Test
Gram 3: Significant difference values ​​should be additionally indicated on the plot, post-analysis Manwitney needs to be indicated
Table 3: It is necessary to check whether the statistical interpretation is correct. How was the random analysis performed and how the variables were set. Methodological review required
general review
Based on the number of samples in the study, this study can be considered as a basic study. A biopsy is an important factor in confirming the disease. Interpretation of this biopsy should suggest any clinical usefulness. In addition, discussion proposals such as the originality of this study or supporting existing evidence should be presented. It looks like it needs improvement.
Author Response

(The authors gave the same response as above.)

Reviewer 3 Report
Dear Authors
I must congratulate you on the inspired work you did.
E-smoking is a relatively now habit, and little we know about the dangers associated with its use.
Your work is an important one towards that direction.
The only thing that needs to be included, is the actual p-value of smoking vs e-cigarettes (group A vs B), especially if it is between 0.05 and 0.1. This level of significance is called suggestive, and it is important in showing trends
Looking forward to your reply
Author Response

(The authors gave the same response as above.)

Round 2
Reviewer 2 Report
I agree to the publication of the improved manuscript.
Congratulations.